# Adherence to the Mediterranean Diet in Women and Reproductive Health across the Lifespan: A Narrative Review

**DOI:** 10.3390/nu15092131

**Published:** 2023-04-28

**Authors:** Maria Karolina Szmidt, Dominika Granda, Dawid Madej, Ewa Sicinska, Joanna Kaluza

**Affiliations:** 1Department of Human Nutrition, Institute of Human Nutrition Sciences, Warsaw University of Life Sciences (WULS-SGGW), Nowoursynowska 166, 02-787 Warsaw, Poland; dawid_madej@sggw.edu.pl (D.M.); ewa_sicinska@sggw.edu.pl (E.S.); 2Department of Nutrition Physiology and Dietetics, Institute of Sport-National Research Institute, Trylogii 2/16, 01-982 Warsaw, Poland; dominika.granda@insp.pl

**Keywords:** Mediterranean diet, reproductive health, women, menstrual cycle, sexual dysfunction, polycystic ovarian syndrome, endometriosis, infertility

## Abstract

The Mediterranean diet (MD) has been previously proven to have various health-related benefits; however, its effect on women’s reproductive health over a lifespan is yet to be summarized. This study aimed to summarize the evidence-based knowledge regarding the association between the MD and selected reproductive health outcomes. By searching PubMed, ScienceDirect, and Google Scholar databases, as well as reference lists, 21 studies were included in this narrative review. The available evidence was very limited; however, there is some suggestion that higher adherence to the MD may be positively associated with a lower risk of early age menarche (1 study) and shorter menstrual cycles (1 study), but is unrelated to dysmenorrhea (1 study). Moreover, no study to date has examined the relationship between the MD and the onset age of natural menopause. Considering reproductive health diseases, there was limited evidence that a higher adherence to the MD was associated with a lower risk of premenstrual syndrome (1 study), an improvement in sexual health and a lower prevalence of sexual dysfunction (3 studies), and an improvement in the general condition of with endometriosis and the pain they can experience (1 study). The largest number of studies were found for polycystic ovarian syndrome (PCOS; 7 studies) and infertility (6 studies). Results showed that a higher adherence to the MD was associated with a lower risk of infertility, while results for PCOS were unclear, but mostly indicated a beneficial trend. Further investigations are necessary to establish the role of adherence to the MD in reproductive health maintenance and its possible role in the prevention and treatment of reproductive health diseases in women.

## 1. Introduction

As mentioned by the World Health Organization (WHO), due to various biological and gender differences between men and women, sex has a significant impact on health [1]. Women have a longer life expectancy than men in most countries around the world, yet they experience more disabilities, comorbidities, and poorer mental health [2]. Additionally, throughout their lives, women face numerous hormonal transitions related to puberty (e.g., menarche), which are connected with regular menstrual cycles, pregnancy and lactation, and menopause. Each of the listed phases of a woman’s life is considered physiological, but these hormonal transitions put women at an increased risk of psychological and somatic distress [3]. For instance, menopause is associated with a higher risk of cardiovascular disease (CVD) or osteoporosis [4]. Additionally, the female reproductive system is associated with a plethora of diseases, such as premenstrual syndrome, dysmenorrhea, sexual dysfunction, endometriosis, polycystic ovarian syndrome (PCOS), and infertility. Some of these adverse health outcomes and diseases can be reduced by modifiable aspects of lifestyle, such as the observance of a healthy diet.

The Mediterranean diet (MD) has been previously studied and proven to have various health-related benefits, including the prevention of CVD [5], cancers [6], and type 2 diabetes [7]. However, until now, the impact of the MD on women’s reproductive health across their lifespans has not been summarized. Therefore, the aim of this narrative review was to summarize the results from studies investigating the relationship between an adherence to the MD and various aspects of women’s health across their lifespan.

## 2. Methods

### 2.1. Search Strategy

A narrative review of the literature was conducted following the Academy of Nutrition and Dietetics Narrative Review Checklist (ANDJ) [8]. PubMed, ScienceDirect and Google Scholar databases were searched from inception to March 20, 2023. To identify potential studies, we used a combination of the following terms: “Mediterranean diet” with “menstrual cycle”, “menarche”, “menstruation”, “premenstrual syndrome”, “PMS”, “premenstrual dysphoric disorder”, “PMDD”, “menopause”, “polycystic ovary syndrome”, “PCOS”, “endometriosis”, “infertility”, “fertility”, “sexual health”, “female sexual dysfunction”, and “female sexual function index”. In addition, a manual search of the reference lists of the identified publications was conducted to identify any additional potentially eligible studies.

### 2.2. Inclusion and Exclusion Criteria

The inclusion criteria consisted of the following: (1) a case-control, cross-sectional, prospective cohort, experimental, and meta-analysis study design; (2) studies in which the MD was identified; (3) studies that involved women; and (4) articles written in English. The exclusion criteria consisted of the following: (1) animal studies; and (2) studies lacking a definition of the MD.

### 2.3. Study Selection and Data Extraction

The titles and abstracts of the identified studies were screened to determine their eligibility for inclusion in the review. Subsequently, a comprehensive evaluation of the full-text articles was performed based on the predefined inclusion and exclusion criteria. In the event of doubts regarding the inclusion of a publication in the review, the eligibility of an article was discussed among co-authors, and the final decision was made by the senior author (J.K.). Efforts were made to obtain any missing information from potentially eligible studies by contacting the authors.

### 2.4. Quality Assessment

The quality of the studies was assessed using the Newcastle–Ottawa Quality Assessment Scale (NOS) for case-control, cohort, and cross-sectional studies, while the Critical Appraisal Skills Programme (CASP) checklist was used for randomized controlled trials [8,9]. The NOS is endorsed by the Cochrane Collaboration as a recommended tool for assessing the quality of non-randomized studies [10]. It evaluates studies based on three pre-defined criteria: selection, comparability, and exposure/outcome. The quality assessment was carried out independently by two co-authors (M.S., D.G.), and in the event of any disagreement, the senior author (J.K.) made the final decision. Detailed scoring and thresholds, according to which studies were categorized by the Agency and Healthcare Research and Quality (AHQR) standards, are shown in Appendix A (Appendix A). Briefly, case-control studies with scores of 0–3 were viewed as low quality, scores of 4–6 as medium quality, and scores of 7–9 as high quality. As for cross-sectional studies, the scores ranged from 0 to 3 points for low-quality studies, 4–6 points for medium-quality studies, and 7–10 points for high-quality studies. Cohort studies that received “0 or 1 point in the selection domain” or “0 points in the comparability domain” or “0 or 1 point in the outcome domain” were rated as low quality. Studies with “2 points in the selection domain” or “1 or 2 points in the comparability domain” or “2 or 3 points in the outcome domain” were rated as medium quality, while those with “3 or 4 points in the selection domain” or “1 or 2 points in the comparability domain” or “2 or 3 points in the outcome domain” were classified as high quality. Randomized controlled trials with “no or few limitations” were evaluated when the assessments for most items were “yes”. The term “minor limitations” applied to trials in which the assessments for most items were “yes” or “cannot tell”, while the term “major limitations” applied to trials in which the assessments for one or more questions were “no”.

## 3. Reproductive Health

Until now, only a few studies have provided results regarding the relationship between an adherence to the MD and the physiological aspects of reproductive health, such as the age of menarche or menstrual characteristics. In Figure 1 and Table 1, we present a summary of the included studies results.

### 3.1. Menarche

Menarche is the first menstrual period experienced by girls, typically occurring between the ages of 11 and 14, although the age range can vary [11]. The timing of menarche can be influenced by various factors, including genetics (e.g., maternal age at menarche), socioeconomic factors (e.g., urban/rural residence, parental education, economic status) and lifestyle factors (e.g., diet, physical activity) [12], and may have broad implications for later health [13]. An early age of menarche is associated with a higher risk of diabetes [14], metabolic syndrome, myocardial infarction, ischemic stroke [15], and some types of cancers (e.g., endometrial, breast, and liver) [16]. On the other hand, the late timing of pubertal events is associated with a higher risk of osteoporosis [17]. Therefore, recent trends that see an earlier age of menarche have prompted researchers to investigate whether diet could affect menarche age.

#### 3.1.1. Observational Studies

Only a single longitudinal cohort study of low quality has examined the association between the MD and the age of menarche [18]. The study group consisted of 202 girls aged 9 or 10 at baseline (Table 1). It was found that girls with a higher adherence to the MD (measured by an adapted Mediterranean-like Diet score; range 0–9 points) had a 55% (95% CI: 29–72%, *p* < 0.01) lower risk of experiencing menarche at an early age than girls with a lower adherence. Each increase in the score regarding adherence to the MD was associated with an 11% lower risk of experiencing an earlier menarche (95% CI: 2–18%, *p* < 0.05). Further analysis showed that a low, compared to high non-fat or low-fat, dairy product consumption was associated with a 37% risk reduction (95% CI: 9–57%, *p* < 0.05) of experiencing menarche at an earlier age. In addition, high vs. low vegetable consumption was associated with a 34% (95% CI: 5–54%, *p* < 0.05) lower risk of an earlier menarche. For other components of the MD, no significant associations were found.

In other studies on the components of the MD and the age of menarche, it was found that nuts and vegetable oils are associated with experiencing menarche at a later age [19], while meat and dairy consumption are associated with an earlier menarche age [20,21,22,23]. Results regarding dairy consumption and the age of menarche are inconsistent [20,24].

#### 3.1.2. Possible Mechanism

The mechanism by which the MD may influence menarche is not clear, but it is considered to be related to the association between MD adherence and the level of sex hormone-binding globulin (SHBG) and endogenous estrogens in women, which may result in a later puberty [18,25,26]. Furthermore, it could be related to the anti-inflammatory and antioxidant properties of the diet [27,28]. It is considered that these properties may regulate hormonal balance and reduce inflammation, which could contribute to delaying the onset of menarche.

### 3.2. Menstrual Cycle

The menstrual cycle is the natural biological process that occurs in the female reproductive system that makes pregnancy possible. It involves a series of hormonal and physiological changes that occur over an average period of 28 days (ranging from 21 to 35 days) [11]. The menstrual cycle can be divided into several phases: the follicular, ovulation, and luteal. During the follicular phase, the ovary prepares to release an egg, which occurs during ovulation. The luteal phase begins after ovulation and lasts until the next menstrual period begins. The usual duration of menstruation is 3–5 days, and the amount of blood loss can range from 10 to 80 mL (average 30 mL). Loss of more than 80 mL is considered abnormal.

The menstrual cycle is an important indicator of reproductive health. Changes in the length of the menstrual cycle, length of menstrual bleeding, and amounts of menstrual flow may indicate underlying health issues or hormonal imbalances [29]. Various factors can affect the characteristics of the menstrual cycle, including lifestyle (smoking, stress, diet, and exercise) or environmental factors.

#### Observational Studies

Only one cross-sectional study of fair medium quality has examined the association between the MD and diverse aspects of the menstrual cycle [30]. In the study of 311 students (21.2 ± 2.6 years), the association between an adherence to the MD (assessed using the KIDMED questionnaire; range 0–16 points) and the duration of the menstrual cycle was observed (Table 1). Women with a low adherence to the MD, in comparison to those with a high adherence, had longer menstrual cycles (33.5 ± 16.2 vs. 30.2 ± 6.3 days, *p* = 0.008); however, no associations were observed regarding the regularity of menstrual cycles, the amount of menstrual flow, duration of menses, or menstrual pain.

No other significant results regarding the correlation between MD components and menstrual cycle characteristics were found.

### 3.3. Menopause

Menopause is the first day after the last women’s menstrual bleeding, connected with the transition between reproductive and post-reproductive periods [31]. This physiological process usually concerns women between 45 and 55 years old, with a median age of 50–52 years among Caucasian women from industrialized countries [32]. The hormonal changes (especially low estrogen and progesterone production) associated with the natural menopausal transition are connected to many symptoms that determine the quality of life and can affect women’s well-being. Common symptoms during the menopausal transition include unregular menstrual cycles and menstrual flow, hot flashes, night sweats, sleeping disorders, emotional instability and mood changes, vaginal dryness, pain during sexual intercourse, and incontinence [31]. During the postmenopausal period, some health problems may develop, such as CVD, hypertension, osteoporosis, etc. However, not all symptoms occur in all women, and their severity can differ between them.

#### 3.3.1. Age of Onset of Menopause

The age at which women experience natural menopause may be affected by many determinants, including sociodemographic, lifestyle, and health factors [33,34]. Factors that include a younger age at menarche, nulliparity, smoking, and very high physical activity may accelerate the onset of menopause. In contrast, a higher education level, better economic status, moderate physical activity, parity, the previous use of oral contraceptive pills, overweight and obesity, and the intake of alcohol in moderation may delay the onset of menopause [33,34].

To the best of our knowledge, until now, no studies have examined the age at which women experience natural menopause in relation to an adherence to the MD. Some specific MD components in relation to women’s age of natural menopause and reproductive lifespan have been examined; however, the number of studies is limited and the evidence remains controversial [35]. For example, in one study, the high consumption of green and yellow vegetables (but not others) was associated with a higher age of menopause [36]; in another study, high vegetable consumption was related to a lower age of menopause [37], but another found no association [38]. In addition, the results obtained for the consumption of soya, cereal products, and red meat are inconsistent [37].
nutrients-15-02131-t001_Table 1Table 1Summary of studies investigating the MD and reproductive health outcomes.Authors (Country)Type of StudyNumber and Age of ParticipantsAssessment of Mediterranean DietEffectQuality Assessment ^a^AGE OF MENARCHESzamreta et al.[18] (USA) Longitudinal cohort study*n* = 20210.0 ± 0.58 yearsAdaptedMD scoreNo associationLowMENSTRUAL CYCLEOnieva-Zafra et al.[30] (Spain) Cross-sectional*n* = 31121.2 ± 2.6 yearsKIDMED questionnaireWomen with low adherence had longer menstrual cycles.No association with regularity, amount of flow, duration of menses, or menstrual pain.MediumAGE OF ONSET OF MENOPAUSENo study was conductedKIDMED—Mediterranean Diet Quality Index for children and adolescents. ^a^ Quality assessment for cohort and cross-sectional studies was assessed using the Newcastle–Ottawa Quality Assessment Scale (NOS).


#### 3.3.2. The European Menopause and Andropause Society (EMAS) Position Statement

Based on the results of observational and randomized trials conducted mainly on total adult populations, the EMAS assessed the potential influence and formulated conclusions regarding MD adherence in relation to menopausal health [39]. The EMAS indicated that short-term and long-term adherence to this type of diet is beneficial for women’s health. Short-term adherence to the MD may improve vasomotor function, improve mood, and decrease the risk of depression, while a long-term adherence in peri- and postmenopausal women may reduce all-cause mortality, CVD incidence and mortality, the risk of breast cancer, maintain bone mineral density in healthy women and improve mineral density in women with osteoporosis, and prevent cognitive decline. However, most of these conclusions were based on men’s and women’s studies in a comprehensive range of ages, while a number of studies restricted to perimenopausal women are limited.

#### 3.3.3. Symptoms and Health Problems Related to Perimenopausal Age

In addition to the EMAS statement, several studies have assessed the association between an adherence to the MD and vasomotor symptoms and health problems that mainly develop during the peri- and postmenopausal periods. It is worth noting that some of these studies were conducted among women within a wide range of ages, from 40 to 75 years. Therefore, in this review, we focused only on studies that were conducted among women of perimenopausal age.

In a prospective cohort study of 6040 women (50–55 years of old, from the Australian Longitudinal Study on Women’s Health, followed up over 9 years), it was found that adherence to the MD was inversely associated with vasomotor menopause symptoms, such as hot flushes and night sweats; the multivariate-adjusted OR in women in the highest quintile of adherence to the MD compared to those in the lowest quintile was 0.80 (95% CI: 0.69–0.92, *p* = 0.0004) [40]. In contrast, in a cross-sectional study of 172 women (45–60 years old, recruited from the FLAMENCO study), there were no associations found between the MD and vasomotor functions (assessed via the Kupperman Menopausal Index and the Menopause and Health subscale of the Cervantes Scale) [41].

Moreover, based on a population-based cross-sectional study of Spanish perimenopausal women (*n* = 3508, 48.9 ± 4.0 years), it was found that a high level of adherence to the MD was inversely associated with overweight and obesity, as well as with the occurrence of health symptoms that are typical for menopausal transition (measured using the Menopause and Health subscale of the Cervantes Scale) [42].

In the cross-sectional study (FLAMENCO project, women aged 45–60 years), the cardioprotective influence of the MD in perimenopausal women was observed [41,43]. Women with high a adherence to the MD compared to those with a low adherence had statistically and significantly lower plasma concentrations of total cholesterol, low-density lipoprotein cholesterol (LDL-C), and triglycerides, C-reactive protein, and a lower resting heart rate [43]. Moreover, a multivariable-adjusted statistically positive correlation was found between the MD score and the ratio of gynecoid to total fat mass, and an inverse correlation was found between the MD score and the ratio of android to total fat mass [41].

## 4. Reproductive Health Dysfunction

To date, a limited number of studies have focused on establishing a potential association between an adherence to the MD and reproductive health dysfunctions, such as PMS, dysmenorrhea, sexual dysfunction, endometriosis, PCOS, and infertility. The results of the included studies are presented in Figure 2 and Table 2.

### 4.1. Premenstrual Syndrome

Premenstrual syndrome (PMS) is defined as a combination of physical and psychological symptoms, which occur only in the luteal phase of the menstrual cycle in approximately 30–40% of women of reproductive age [44]. PMS is included in the International Statistical Classification of Diseases and Related Health Problems (ICD-11) with code GA34.40 [45]. There is some disagreement among experts in terms of the diagnostic criteria for PMS; however, most often, it is assumed that reporting at least one symptom occurring in the luteal phase over two or more menstrual cycles indicates PMS. To talk about PMS, the symptoms must significantly affect an individual’s quality of life, and several other conditions must be ruled out (e.g., thyroid diseases, anxiety, depression). There is also a more severe form of PMS, which is called premenstrual dysphoric disorder (PMDD); this involves the same symptoms as PMS, but the symptoms are more severe and often prevent normal functioning. It is estimated that PMDD affects 3–8% of women of reproductive age [46]. Despite extensive research, the mechanism underlying PMS is still unknown; therefore, the available pharmacological treatments (selective serotonin reuptake inhibitors, oral contraception) focus on lowering the severity of symptoms, but do not treat the causes of PMS. Additionally, not all women may decide to undertake the pharmacological treatment due to possible adverse effects and health contraindications [47]. This, therefore, underlines the need for studying modifiable lifestyle elements that could ease or eliminate these symptoms, and diet is one of them.

#### 4.1.1. Observational Studies

So far, only one medium-quality study that evaluates the impact of MD adherence on the occurrence of PMS symptoms has been published (Table 2) [48]. In a group of 262 Korean women aged 20–49 years old, it was shown that a low adherence to the MD (assessed via a modified version of the Mediterranean Diet Adherence Screener) was associated with an increased risk of PMS. The proportion of women with PMS was significantly lower in the highest tertile of adherence to the MD compared to the lowest (55.4% vs. 74.4%, *p* = 0.045). Previously, a higher adherence to a Mediterranean-style diet had been proven to be beneficial to maintaining well-being in various groups of adults [49,50,51,52], and decreased mood is a common symptom of PMS.

More research has been conducted regarding individual components of the MD in the context of PMS. The role of fruit and vegetable intake in the occurrence of PMS has been investigated in three studies [53,54,55]. In Iranian nursing students (156 with and 151 without PMS), it was found that a higher intake of vegetables, especially cruciferous vegetables, was associated with a significantly lower risk of PMS [55]. On the contrary, in two other studies, no association was observed between fruit and vegetable intake, and the risk of PMS [54,56]. One study examined the relationship between fish consumption and PMS in a group of athletes compared to women with regular physical characteristics [57]. Fish consumption was shown to be associated with a decreased risk of underperformance in the athlete group (*n* = 200); however, a similar association was not found in the non-athlete group (*n* = 112). It is worth mentioning that the study was conducted among the Japanese population, which is characterized by a high fish and seafood consumption.

#### 4.1.2. Possible Mechanism

The protective properties of the MD on PMS are presumably connected with a high content of omega-3 fatty acids. This was proven to provide relief from the severity of PMS symptoms, possibly due to its ability to activate G-protein-coupled receptor 40, which further leads to the release of beta-endorphin into the hypothalamus and results in a reduction in pain and depressive-like symptoms of PMS [58].

### 4.2. Dysmenorrhea

Dysmenorrhea is a common gynecological condition that affects many women during their reproductive years [59]. It is characterized by painful menstrual cramps of uterine origin, and according to ICD-11 (ICD-11: GA34.3.), it can be categorized into primary (lack of organic disease) and secondary dysmenorrhea (associated with an identifiable disease, e.g., endometriosis) [45,59,60]. The most severe pain is observed during the first or second day of menstrual bleeding, and may be accompanied by other symptoms (e.g., diarrhea, nausea, headaches, backaches, and others). While the exact cause of dysmenorrhea is not fully understood, research suggests that lifestyle factors, including diet, may play a significant role [59,60,61,62].

#### Observational Studies

Only one cross-sectional study of medium quality has examined the association between the MD and primary dysmenorrhea (Table 2) [30]. The results of the study of 311 health science students (21.2 ± 2.6 years) indicated that there is no statistically significant association between the level of adherence to the MD (assessed using the KIDMED questionnaire, range 0–16 points) and the intensity of menstrual pain when considered as a continuous variable, nor with intensity categories (mild, moderate, severe). However, an analysis of the components of the MD showed that almost 2-fold more women without dysmenorrhea consumed a second serving of fruit compared to women with dysmenorrhea (respectively: 48.6% vs. 28.3%, *p* = 0.04). Women who did not consume a second serving of fruit daily had a 34% (95% CI: 0.16–0.72) higher odds ratio of experiencing primary dysmenorrhea than women who fulfilled this recommendation. In addition, women who consumed pulses ≤1 time/week had a 2.3-fold (95% CI: 1.01–5.35) higher odds ratio of experiencing primary dysmenorrhea compared to women who consumed them > 1 time/week. For other components of the MD, no significant associations were found.

The results of other studies have shown that such components of the MD, such as vegetables, fruits, and legumes, are associated with a lower risk of primary dysmenorrhea [30,63,64].

No studies on the association between an adherence to the MD and secondary dysmenorrhea were found.

### 4.3. Sexual Dysfunction

The WHO defines sexual health as a state of physical, emotional, mental, and social well-being in relation to sexuality. This state requires a positive and respectful approach to sexuality and sexual relationships, while experiencing pleasurable and safe sexual experiences that should be free from coercion, discrimination, and violence [65]. One of the common problems associated with sexual health is female sexual dysfunction (FSD; ICD-11: HA40), which, according to certain studies, affects about 40–43% of women [45,66,67]. These values may be underestimated because some women do not report problems with sexual dysfunction, which is characterized by a disorder related to experiencing sexual pleasure, orgasm, sexual desire, or pain during intercourse [68,69]. A commonly used measure of FSD is the Female Sexual Function Index (FSFI), which includes an assessment of six domains: desire, arousal, lubrication, orgasm, satisfaction, and pain [70].

Many studies have demonstrated the connection between metabolic syndrome and FSD. An increase in female sexual health disorders has been observed in those who experience obesity [71], hypertension [72], diabetes [73], dyslipidemia [74], or the cumulative burden of cardiovascular risk [75]. Therefore, a lifestyle that improves health and reduces the components of metabolic syndrome can positively affect sexual function [65]. It has been confirmed that the MD has a beneficial effect on FSD in women with metabolic syndrome, so it seems that a lifestyle based on this dietary pattern may be a beneficial strategy for improving women’s sexual health [76].

#### 4.3.1. Experimental and Observational Studies

A randomized trial study with major limitations, which included 59 women with metabolic syndrome, assessed the association between the MD and FSD (Table 2) [76]. After two years of adherence to the MD, the intervention group (higher intake of fruits, vegetables, nuts, whole grain products, and olive oil) had a statistically significant higher FSFI compared to the control group (19.7 ± 3.1 vs. 26.1 ± 4.1 points, *p* = 0.01), which was a measure of the improvement in the sexual health of women. At the same time, a reduction in serum CRP levels was observed, though none of the FSD domains (i.e., desire, arousal, lubrication, orgasm, satisfaction, pain) improved significantly. However, in an 8-year follow-up study (with major limitations) of the same group, women with type 2 diabetes in the MD group experienced a lower level of sexual health deterioration (lower reduction in FSFI) than those in the low-fat group [77].

A cross-sectional study (medium-quality) of 595 diabetic women also showed an association between an adherence to the MD and a reduction in FSD. The women in the highest tertile of adherence to the MD had a lower prevalence of sexual dysfunction compared to those in the lowest and middle tertiles (47.6% vs. 53.9% and 57.8%, respectively, *p* = 0.01) [78].

#### 4.3.2. Possible Mechanism

The mechanism by which a Mediterranean-style diet can improve FSD is not clear. However, the observed association between sexual health, improved metabolic syndrome components, and the MD may suggest the significant role of antioxidant components. The pro-inflammatory state developed as a result of chronic oxidative stress is one of the causes of the development of conditions that are categorized as metabolic syndrome [76,79].

### 4.4. Endometriosis

Endometriosis is a chronic inflammatory disease in which tissue resembling the endometrium (the lining of the uterus) grows outside the uterus, causing pelvic pain, scarring, and/or infertility (ICD-11: GA10) [45]. It is estimated that this condition affects roughly 10% of reproductive-age women globally [80]. Most cases of this disease occur in women between menarche and menopause, with a peak incidence between 25 and 45 years of age. The symptoms associated with endometriosis include severe life-impacting pain during periods, sexual intercourse, bowel movements and urination, abdominal bloating and nausea, fatigue, depression, or anxiety. It is estimated that 30% to 50% of women with endometriosis experience infertility. Risk factors for endometriosis include, among others, early menarche (before the age of 11), short genital cycles (lasting less than 27 days), Caucasian race, and a low Body Mass Index (BMI) [81].

Endometriosis is an estrogen-dependent disease, so therapeutic strategies focus primarily on hormonal treatment. Some widely employed therapies include non-steroidal anti-inflammatory medications and analgesics (painkillers). An alternative treatment option is the surgical removal of endometrioid lesions; however, the recurrence rate is up to 50% within five years of surgery. An important factor in the primary prevention of endometriosis is the maintenance of an appropriate lifestyle, including a healthy diet and physical activity [82].

#### 4.4.1. Experimental Studies

Only one experimental study with major limitations examined the influence of the MD on endometriosis-associated pain (Table 2) [83]. In the study, of the 68 women with a previous laparoscopic diagnosis of endometriosis and postoperative endometriosis-associated pain, 43 declared their adherence to the nutritional regimen throughout the 5-month experiment. Using a Numeric Rating Scale (NRS), a decrease in general pain (NRS 4.2 ± 3.0 vs. 2.0 ± 2.3, *p* < 0.01), as well as an improvement in the general condition (NRS 6.7 ± 2.2 vs. 8.5 ± 1.7, *p* < 0.01), were observed. However, this study has several limitations, such as the lack of a control group, self-reported adherence to the MD, and a lack of regular meetings with a dietician. It is, therefore, possible that the pain relief was not just the result of a dietary change towards the MD, but a placebo effect or the influence of other lifestyle factors (e.g., increased physical activity).

Furthermore, in a recently published review paper [82], it was found that a higher intake of specific MD components, such as vegetables, fruit, and long-chain *n*-3 fatty acids, may be beneficial in reducing the risk of endometriosis, while the consumption of trans fats and a higher intake of red meat may increase the risk.

#### 4.4.2. Possible Mechanism

Fish and cold-pressed oils have been shown to exert anti-inflammatory effects. In particular, extra virgin olive oil, which contains the substance oleocanthal, displays a similar structure to the molecule ibuprofen and takes effect via the same mechanism, i.e., cyclooxygenase inhibition. In addition, the increased amount of dietary fiber in the diet provides a eupeptic effect, whereas foods high in magnesium could prevent an increase in the intracellular calcium level, which is essential for muscular contraction and thereby, might decrease chronic pelvic pain [83].

### 4.5. Polycystic Ovary Syndrome

PCOS is a condition defined by the presence of at least two of the following three criteria: oligoovulation or anovulation, signs of clinical or biochemical hyperandrogenism, and polycystic ovarian morphology after the exclusion of secondary causes (ICD-11: 5A80.1) [45,84]. The prevalence of PCOS worldwide ranges from 4% to 21% among reproductive-age women, depending on the diagnostic criteria used [84]. The syndrome is associated with reproductive features (including hyperandrogenism, lack of ovulation, irregular periods, and infertility), metabolic features (increased risk of impaired glucose tolerance, type 2 diabetes, and CVD), and psychological features (increased risk of depression and anxiety) [85]. Patients with PCOS were found to have an increased prevalence of obesity [86]. First-line management of PCOS includes lifestyle interventions that focus on weight loss and dietary modifications, with efforts to improve insulin sensitivity and prevent long-term health consequences [85].

#### 4.5.1. Randomized Controlled Trials

Two randomized controlled trials were conducted in overweight or obese women suffering from PCOS (Table 2) [87,88]. The studies aimed to evaluate the possible beneficial effects of energy-restricted dietary models on anthropometric, metabolic, and endocrine parameters: a hypocaloric MD vs. a ketogenic diet [87], and a MD combined with a low-carbohydrate vs. low-fat diet [88].

In the study by Cincione et al. [87], which had major limitations, participants followed either the moderately hypocaloric MD (*n* = 71, BMI 33.6 ± 4.9 kg/m^2^, energy intake 500 kcal lower than the patients’ daily requirements) or the very-low-calorie ketogenic diet (*n* = 73, BMI 33.4 ± 5.7 kg/m^2^, energy intake at around 600 kcal/day) for a short period (45 days). After interventions in both groups, significant changes in the anthropometric and biochemical parameters were observed, with a higher improvement observed in those on the ketogenic diet (BMI: −4.15 ± 1.31 vs. −1.17 ± 0.61 kg/m^2^, *p* < 0.001; insulin resistance index: −5.70 ± 3.94 vs. −1.90 ± 1.97, *p* < 0.001; total testosterone: −7.40 ± 4.01 vs. −5.30 ± 4.07 ng/dL, *p* < 0.001; and luteinizing hormone: −5.51 ± 3.23 vs. −3.07 ± 1.80 mUI/mL, *p* < 0.001). Moreover, after the intervention, in the ketogenic diet group, some patients (34%) experienced a natural reappearance of a regular menstrual cycle after years of amenorrhea. However, the study was limited in its comparison of two diets that have disparate energy intake levels, making it impossible to determine whether the diets had a considerable effect on the metabolic and hormonal parameters of the participants.

Mai et al. [88] studied the therapeutic effect of the MD combined with a low-carbohydrate (*n* = 30, BMI 29.4 ± 2.2 kg/m^2,^ a daily carbohydrate intake of 100 g or less) versus low-fat diet model (*n* = 29, BMI 29.6 ± 2.5 kg/m^2^, <40 g of fat intake with up to 10% energy from saturated fats) in overweight women with PCOS for 12 weeks. This study, which had major limitations, showed that both dietary models were effective in modifying anthropometric parameters, and that the metabolic and endocrine parameters in patients undertaking the MD/low-carbohydrate diet were affected to a greater extent. The MD/low-carbohydrate group, compared with the MD/low-fat group, experienced a greater improvement in anthropometric, as well as metabolic and endocrine, parameters (BMI: −2.12 ± 0.57 vs. −1.78 ± 0.36 kg/m^2^, *p* < 0.05; low-density lipoprotein cholesterol: 0.73 ± 0.76 vs. −0.41 ± 1.05 mmol, *p* < 0.05; total testosterone: −0.20 ± 0.24 vs. 0.08 ± 0.11 ng/mL, *p* < 0.001; and luteinizing hormone: −5.28 ± 3.31 vs. −3.39 ± 3.64 mIU/mL, *p* < 0.05). In addition, after the intervention, the recovery of menstrual cycles was observed in a similar number of patients in both groups (87% and 72% of patients, respectively). The authors of that study recommended the MD/low-carbohydrate dietary model in the treatment of patients with overweight-related PCOS.

#### 4.5.2. Case-Control Studies

Two case-control studies have analyzed the specific dietary patterns associated with PCOS (Table 2) [89,90]. The results of the first study, which was of medium quality, indicated that the MD pattern was associated with a lower risk PCOS in women diagnosed with PCOS (*n* = 202) compared to the control group (*n* = 325), with an odds ratio of 0.76 (95% CI: 0.62–0.92) [89]. Moreover, the MD was inversely correlated with the dietary inflammatory index (r = −0.72).

The results of the second study, which was of high quality and conducted using a sample of 276 Spanish women (121 cases and 155 controls), indicated that there are few significant associations between an adherence to the MD (assessed via relative MD score and alternate MD score) and PCOS; the ORs between the extreme quartiles were 1.6 (95% CI: 0.7–3.9) for the relative MD score and 0.8 (95% CI: 0.4–2.0) for the alternate MD score [90].

#### 4.5.3. Cross-Sectional Studies

Three high-quality cross-sectional studies have analyzed the correlation between the MD and women’s PCOS status, but their results are inconsistent (Table 2).

In the cohort of the Australian Longitudinal Study on Women’s Health [85], adherence to the MD dietary pattern, in relation to PCOS diagnosis, was studied over 3 years. It was found that women with PCOS (*n* = 414), compared to those without PCOS (*n* = 7155), were more likely to adhere to the MD (OR = 1.26; 95% CI: 1.15–1.39). This study may indicate that there is an improvement in the quality of women’s diets following a diagnosis of PCOS.

In another cross-sectional study, the association between adherence to the MD and the clinical severity of PCOS in a cohort of Spanish treatment-naïve women with PCOS (*n* = 112, BMI 31.0 ± 5.7 kg/m^2^) was examined and compared to the BMI-matched control group (*n* = 112, BMI 30.8 ± 5.6 kg/m^2^) [91].It was found that women with PCOS had lower MD scores (6.97 ± 2.72 vs. 8.12 ± 2.80, *p* < 0.001). The PCOS participants with a low adherence to the MD (≤5 vs. ≥10 scores), compared to those with a high adherence, had a higher CRP concentration (1.7 ± 1.0 vs. 0.3 ± 0.2 ng/dL, *p* = 004), higher testosterone levels (36.9 ± 10.3 vs. 20.8 ± 0.37 ng/dL, *p* < 0.001), and a higher Ferriman–Gallwey score (24.5 ± 6.8 vs. 4.6 ± 9.7, *p* < 0.001). Moreover, based on a subgroup of 94 women with obesity (BMI 38.2 ± 6.6 kg/m^2^), cardio-metabolic risk factors were evaluated [92]. The participants were classified according to the metabolic syndrome criteria into two groups: metabolically healthy obese (MHO; *n* = 54) and metabolically unhealthy obese (MUO; *n* = 40). The study showed that PCOS/MUO women had worse endocrine and metabolic profiles. None of the study participants had a high adherence to the MD, but it was found that being PCOS/MUO was associated with a lower MD adherence (OR = 0.28, 95% CI: 0.17–0.45). Furthermore, review papers [93,94] have shown that the key components of the MD that promote health among PCOS patients are extra-virgin olive oil, *n*-3 polyunsaturated fatty acids, polyphenols, and dietary fiber.

#### 4.5.4. Possible Mechanism

It is likely that MD improves the condition of women with PCOS by reducing inflammatory and oxidative stress markers and improving the lipid profile, insulin sensitivity, endothelial function, and anti-atherosclerotic and anti-thrombotic properties. The exact mechanisms of inflammation in PCOS are not yet fully understood, but this condition is mediated by obesity, insulin resistance, and high androgen levels [93,94].

### 4.6. Infertility

The WHO defines infertility as a failure to become pregnant after 12 months or more of regular unprotected sexual intercourse [45]. The results of many studies acknowledge the growing problem of infertility in men and women. It is estimated that the problem of infertility may affect up to 48 million couples in the world. However, this value may not be accurate because the estimate does not reflect the whole world [95,96,97,98].

The International Statistical Classification of Diseases and Related Health Problems (ICD-11) classifies female infertility with the code GA31.0 [45]. There are several possible causes of this disease. It can be attributed to several physiological abnormalities, including irregularities in the endocrine system and/or abnormalities in the specific organs of the female reproductive system (e.g., endometriosis, abnormal ovarian function, tubal infections). Conversely, idiopathic causes of female infertility can be closely related to various factors, such as overweight or obesity, cigarette smoking, alcohol consumption, adherence to an unhealthy diet that is characterized by a low intake of antioxidant components and a high intake of pro-inflammatory substances, and exposure to synthetic chemicals such as perfluoroalkyl substances [98,99,100,101].

Due to the role of reactive oxygen species (ROS) in placental and embryonic development, an increase in ROS levels is observed during the first week of pregnancy [102,103]. Excess ROS and/or the deficiency of antioxidants can cause oxidative stress. Long-term chronic oxidative stress can cause defects in the functioning of cells in the reproductive system via lipids’ peroxidation of the cell membranes and DNA damage [102,104]. Accordingly, some studies have emphasized the need to assess oxidative stress levels in the diagnosis of infertility and fertilization capacity [95,105].

Most studies confirm the beneficial effects of MD on female fertility. Consequently, the MD may improve the reproductive system’s antioxidant status and support in the treatment of idiopathic infertility.

#### 4.6.1. Observational Studies

Several prospective studies have suggest that the MD may have a beneficial effect on female fertility (Table 2). In a high-quality cohort study involving 590 infertile women before in vitro fertilization (IVF) treatment, a higher adherence to the MD was significantly associated with a higher number of available embryos (higher vs. lower MD score: 8.4 ± 5.3 vs. 7.4 ± 4.7, *p* = 0.028). In addition, adherence to the MD was positively correlated with the number of fertilized oocytes and the embryo yield (r = 0.089, *p* = 0.039 and r = 0.102, *p* = 0.018, respectively) [106].

The results of another prospective study (high quality) of 244 female patients before IVF treatment showed that women in the lowest tertile of MD score, compared with those in the highest tertile (≤30 vs. ≥36), had significantly lower rates of clinical pregnancy (29.1% vs. 50.0%, *p* = 0.01) and live birth (26.6% vs. 48.8%, *p* = 0.01). In women aged <35 years, a 2.7-time higher likelihood of achieving clinical pregnancy and live birth was associated with a higher adherence to the MD [107].

In a prospective study of medium quality, the effect of the MD on the fertility of 357 women who had completed at least one cycle of assisted reproductive technologies (ART) was assessed. It was found that women in the 2nd to 4th quartiles of MD score had a significantly higher probability of achieving live birth (0.44, 95% CI: 0.39–0.49) than women in the first quartile (0.31, 95% CI: 0.25–0.39) [108].

In addition, in another study of medium-quality involving 474 women before IVF treatment, the effects of the MD on embryo transfer, clinical pregnancy, and live birth were evaluated. Women who were older than 35 years with an intermediate MD score had the lowest risk of not achieving a clinical pregnancy (RR = 0.84, 95% CI: 0.71–1.00). A higher adherence to the MD was not associated with in vitro fertilization outcomes [109].

On the other hand, in a high-quality prospective study that involved 11,072 pregnant women (with no history of pregnancy loss and with a reported history of at least one pregnancy during the study period), adherence to the MD was not observed to have an effect on pregnancy loss. When comparing the high adherence to the MD with the low, the RR was 1.05 (95% CI: 0.95–1.17) [110].

Only one high-quality cross-sectional study that involved 161 couples undergoing IVF assessed the impact of the MD on pregnancy success. It was found that women whose diet included foods typical of the MD had a significantly increased probability of becoming pregnant (OR = 1.4, 95% CI: 1.0–1.9). However, adherence to the MD was not observed to have an effect on the fertilization rate (β < 0.01, *p* = 0.31) or embryo quality (β = 0.01, *p* = 0.35) was found [111].

#### 4.6.2. Possible Mechanism

The ROS defense system includes antioxidant enzymes and substances, such as polyphenols, vitamin C, vitamin A, β-carotene, and folate, that are present in the MD [104]. In addition, dietary fiber can reduce oxidative stress by participating in the assimilation of polyphenols and carotenoids in the gut, and modulating the immune system response by positively influencing the gut microbiome [112]. Therefore, dietary modification and the introduction of MD-specific products may improve the antioxidant status of the reproductive tract and significantly increase the amount of essential fatty acids in the body, which are important for conception and fetal development.
nutrients-15-02131-t002_Table 2Table 2Summary of studies on selected reproductive health dysfunctions and adherence to a Mediterranean diet (MD).Authors (Country)Type of StudyNumber and Age of ParticipantsAssessment of Mediterranean DietEffectQuality Assessment ^a^PREMENSTRUAL SYNDROMEKwon et al. [48](South Korea)Cross-sectionalNon-PMS: *n* = 91 33 (26–37) yearsPMS: *n* = 17131 (26–37) yearsMediterranean Diet Adherence ScreenerLow adherence to MD was associated with increased risk of PMS.MediumDYSMENORRHEAOnieva-Zafra et al. [30] (Spain)Cross-sectional*n* = 31121.2 ± 2.6 yearsKIDMED questionnaireNo association.MediumSEXUAL DYSFUNCTIONEsposito et al.[76] (Italy)Randomized controlled trial studyWomen with metabolic syndrome and FSDIntervention: MDfor 2 yearsMD *n* = 2142.3 ± 4.5 yearsCD *n* = 2841.5 ± 3.9 yearsEvaluated by the nutritionist for 24 months (consultations every month)Improved FSFI and reduced CRP levels in the intervention group. No single sexual domain (desire, arousal, lubrication, orgasm, satisfaction, pain) was significantly ameliorated.Major limitationsMaiorino et al.[77] (Italy)Randomized clinical trial studyWomen with type 2 diabetes and FSDIntervention: MDfor 8.1 yearsMD *n* = 5450.9 ± 9.2 yearsLFD *n* = 5551.2 ± 9.3 yearsMD scoreLess deterioration in the sexual health of the intervention group.Major limitationsGiugliano et al. [78](Italy)Cross-sectional studyWomen with type 2 diabetes and FSD*n* = 59557.9 ± 6.7 yearsMD scoreWomen with the highest MD score had lowest prevalence of sexual dysfunction.MediumENDOMETRIOSISOtt et al.[83] (Austria)Experimental studyWomen with endometriosisIntervention: MDfor 5 months*n* = 6835.3 ± 11.2 yearsSelf-reportedby each patientSignificant relief of generalpain and an improvement in the general condition.Major limitationsPOLYCYSTIC OVARY SYNDROMECincione et al. [87] (Italy)Randomized controlled trialOverweight and/or obese women with PCOSIntervention: hypocaloric MD vs. KD for 45 daysMD *n* = 7133.6 ± 4.9 yearsKD *n* = 7333.4 ± 5.7 yearsEvaluated by the nutritionist through counseling every 2 weeks and reinforced by phone calls every 2–3 daysBoth interventions were effective. The improvement in the anthropometric, metabolic, and endocrine parameters was significantly higher in the KD compared to the MD group.Major limitationsMei et al.[88] (China)Randomized controlled trialOverweight womenwith PCOSIntervention: MD/LC vs. LF for 12 weeksMD/LC *n* = 3028.0 ± 5.3 yearsLF *n* = 2928.1 ± 7.1 yearsEvaluated and monitored by the nutritionistBoth dietary models were effective. MED/LC effectiveness was higher than the LF.Major limitationsWang et al.[89] (China)Case-control studyCases: PCOS patientsControls: healthy womenPCOS *n* = 20230.2 ± 3.4 yearsControls *n* = 32531.8 ± 3.8 yearsMD patternProtective association with PCOS.MediumCutillas-Tolin et al.[90] (Spain)Case-control studyCases: PCOS patientsControls: healthy womenPCOS *n* = 121Controls *n* = 15529.1 ± 5.7 yearsRelative MD scoreAdapted MD scoreNo associations.HighMoran et al. [85] (Australia)Population cross-sectional studyWomen with and withoutself-declared PCOSPCOS *n* = 41433.5 ± 0.1 yearsNon-PCOS *n* = 715533.7 ± 0.1 yearsMD patternProtective association MD with PCOS.HighBarrea et al.[91] (Italy)Cross-sectional studyWomen with and withoutdiagnosed PCOSPCOS *n* = 11224.2 ± 5.5 yearsNon-PCOS *n* = 11224.1 ± 5.1 yearsPREDIMED questionnairePCOS vs. non-PCOS group had a lower adherence to the MD.HighBarrea et al.[92] (Italy)Cross-sectional studyTreatment-naïve women with PCOS and obesityPCOS MHO *n* = 5423.8 ± 3.7 yearsPCOS MUO *n* = 4024.5 ± 3.7 yearsPREDIMED questionnaireMUO vs. MHO patients had a lower adherence to the MD.HighINFERTILITYSun et al.[106] (China)Prospective cohort studyInfertile women before IVF treatmentLMD *n* = 36231.85 ± 3.68 yearsHMD *n* = 22831.67 ± 3.80 yearsMD scoreHigher number of available embryos; positively correlated with the number of fertilized oocytes and embryo yielded.HighKarayiannis et al.[107](Greece)Prospective cohort studyWomen before first IVF treatmentT1 *n* = 7935 (32–37) yearsT2 *n* = 7936 (32–39) yearsT3 *n* = 8636 (34–38) yearsMD scoreHigher rates of clinical pregnancy and live birth.HighGaskins et al. [108](United States)Prospective cohort studyWomen with at least one cycle of ART*n* = 35735.3 ± 4.0 yearsMD scoreHigher probability of live birth.MediumRicci et al. [109](Italy)Prospective cohort studyWomen before IVF treatment*n* = 47436.6 ± 3.6 yearsMD scoreProtective effect of intermediate adherence to MD on oocyte number and clinical pregnancy in women > 35 years.MediumGaskins et al. [110] (United States)Prospective cohort studyWomen with no history of pregnancy loss and who reported at least one pregnancyTotal *n* = 11,072Q1 *n* = 235630.0 (28.0–33.0) yearsQ4 *n* = 267732.0 (30.0–35.0) yearsAdapted MD scoreNo effect on the risk of pregnancy loss.HighVujkovic et al. [111] (Netherlands)Cross-sectional studyWomen undergoing IVF treatmentLMD *n* = 5435.2 (23.2–43.7) yearsIMD *n* = 5433.9 (23.7–40.6) yearsHMD *n* = 5337.2 (29.3–42.1) years
MD pattern
Significantly increased probability of pregnancy.HighART—Assisted Reproductive Technologies, CD—control diet, CRP—C-reactive protein, FFQ—Food Frequency Questionnaire, FSD—female sexual dysfunction, FSFI—female sexual function index, HMD—high adherence to the Mediterranean diet, IMD—intermediate adherence to the Mediterranean diet, IVF—in vitro fertilization, KD—ketogenic diet, LF—low-fat diet, LFD—low-fat diet, LMD—low adherence to the Mediterranean diet, MD—Mediterranean diet, MD/LC—Mediterranean diet combined with a low-carbohydrate diet, MHO—metabolically healthy obesity, MUO—metabolically unhealthy obesity, PCOS—polycystic ovary syndrome, PREDIMED—PREvention with MEDiterranean Diet, T—tercile. ^a^ Quality assessment for cohort, case-control and cross-sectional studies was assessed using the Newcastle–Ottawa Quality Assessment Scale (NOS), and the RCT was assessed using the Critical Appraisal Skills Programme (CASP) checklist.


## 5. Beneficial Effects of MD on Women’s Reproductive Health: Impact of Single Components or Synergistic Effect?

While some studies have shown a positive correlation between adherence to the MD and reproductive health, the underlying mechanisms of action are still unclear. To date, very general dependencies are seen for specific components of the MD, but understanding which characteristics of the MD may contribute to these reproductive health benefits is an area of interest. One potential contributing factor is the high intake of fruits, vegetables, whole grains, legumes, nuts and seeds, and olive oil, which are rich in various vitamins, minerals, and phytoactive components that may have protective effects on reproductive health. For example, vitamin E, which is found in olive oil, has been shown to have antioxidant properties that could potentially reduce oxidative stress and inflammation, which may be beneficial in conditions such as dysmenorrhea, endometriosis, and sexual dysfunction [113,114,115,116]. The whole grains present in the MD may also play a role in reproductive health via their high content of dietary fiber, which can help regulate blood sugar levels, reduce inflammation, and promote the growth of desirable gut microbiota that are relevant to conditions such as PMS, menstrual irregularities, and PCOS [117,118,119,120,121]. In addition, the moderate consumption of red wine may have a positive effect on reproductive health outcomes due to resveratrol’s anti-inflammatory and antioxidant properties, while a high intake of fish ensures an adequate supply of omega-3 fatty acids, which are particularly relevant with regard to the prevention of PMS and infertility [58,122]. Additionally, the MD is characterized by a limited intake of animal-origin products, which may contribute to a lower intake of saturated fats and a higher intake of unsaturated fats (such as those found in nuts, seeds, and fish); this may have beneficial effects on reproductive health, especially PCOS and infertility [122,123].

However, it is highly likely that the synergistic effects of the overall MD dietary pattern, rather than isolated components, contribute to its reproductive health benefits; however, the mechanism behind its effects seems complicated and may include gene expression, epigenetic modifications, and signaling pathways [124,125,126]. Furthermore, it is plausible that the impact of the MD conjuncts with other lifestyle factors; for example, people with a high adherence to the MD, compared to those with a low adherence, are characterized by other healthy beneficial factors, such as higher levels of physical activity, non-smoking, or lower stress levels.

## 6. Implications for Future Research

In recent years, the MD has been linked to numerous health benefits, but there are still major gaps in understanding the association between this dietary pattern and women’s reproductive health. Particularly, very limited evidence have been found regarding the menarche age, menstrual cycle, menopause age, and PMS, dysmenorrhea, and endometriosis. Based on this narrative review, we have provided directions for future researchers to fill the gap and increase knowledge in this area.

Currently, not only is there limited research on the relationship between MD and women’s reproductive health, but the results are sometimes contradictory. A positive association between an adherence to the MD and some reproductive health outcomes was shown in 16 out of 21 studies (5 RCT, 5 cross-sectional studies, 4 prospective studies, 1 case-control study, and 1 experimental study). Although it may seem a large number (76.2%), an in-depth analysis of the studies has shown that there is an extensive variety in the quality of the studies (6 studies of high quality, 5 studies of medium quality, and 5 RCT studies with major limitations). Longitudinal studies are needed to better understand the long-term effects of the diet on reproductive health outcomes. In addition, there is a need to perform well-designed randomized controlled trials, which would abolish the placebo aspect and develop specific nutritional recommendations for women with reproductive health problems. Within such research, the ongoing monitoring of interventions is crucial, and recommendations regarding MD adherence should not only be general, but should also address specific aspects of the diet. Most research on adherence to the MD and reproductive health has been conducted in Western populations. To better understand the potential benefits of this diet on reproductive health, researchers need to conduct studies in different populations, distinguishing those within and outside the Mediterranean region. Furthermore, the standardization of the MD in future research is essential in order to ensure the consistency and comparability of results. Although the MD is widely recognized as a healthy dietary pattern, its specific composition may vary across studies or populations, leading to potential inconsistencies in findings. Of the 21 studies included in this review, 11 different approaches were employed in order to define MD adherence, leading to difficulties in pooling and interpreting results. Therefore, standardizing the MD should involve defining a set of criteria or guidelines for its composition, including recommended food groups, portion sizes, and other relevant factors, such as the participants’ place of residence (in or outside the Mediterranean Bay). This would help researchers to precisely and consistently implement the MD in their studies, allowing for the meaningful comparison of results across different research settings.

While it is essential for researchers to explore different aspects of the relationship between the MD and reproductive health (including the underlying mechanisms), it is also crucial that they follow a consistent methodology in order to ensure the validity and reliability of their findings. Accordingly, it is recommended that researchers follow a standardized methodology when conducting studies on the MD and reproductive health. This should include the following: using similar inclusion and exclusion criteria when selecting study participants (e.g., age range, medical history); using similar tools to determine adherence to the MD; collecting dietary data using validated instruments; and controlling for potential confounding variables that may affect the relationship between the MD and reproductive health.

## 7. Conclusions

Based on limited evidence, the present narrative review has shown that an association between adherence to the MD and women’s reproductive health is possible, but further studies are required in order to draw conclusions. To establish whether the MD is associated with overall reproductive health and could be useful in preventing reproductive health diseases, longitudinal and high-quality intervention studies are required; these should have clear case definitions and a homogenous methodology for the assessment of the MD.

## Figures and Tables

**Figure 1 nutrients-15-02131-f001:**
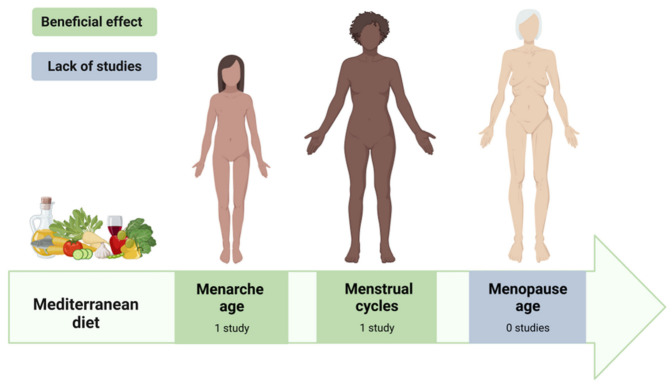
Summary of study results regarding the association between an adherence to the MD and women’s reproductive health (Created with BioRender.com, accessed on 5 April 2023).

**Figure 2 nutrients-15-02131-f002:**
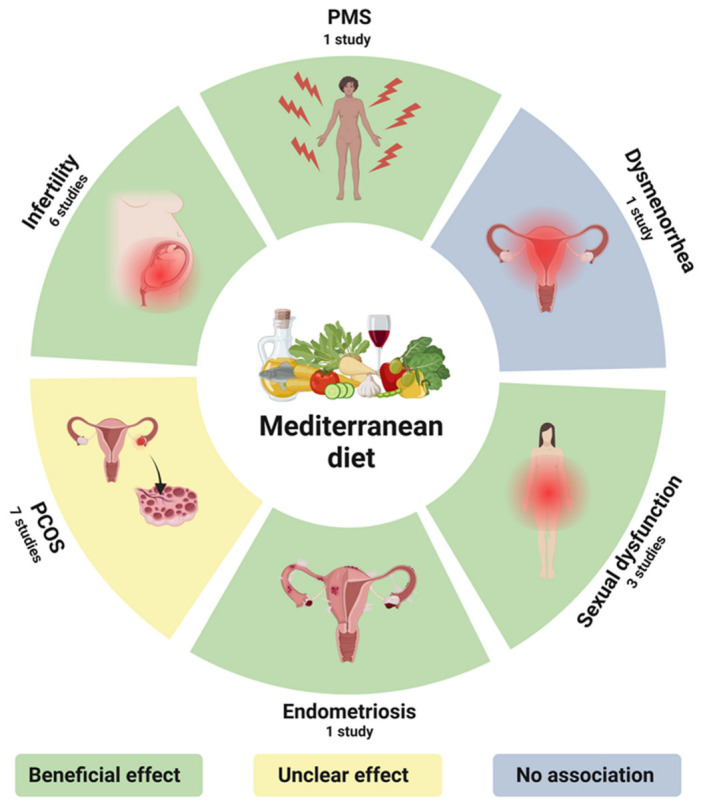
Summary of study results regarding the association between an adherence to the MD and women’s reproductive health dysfunctions (Created with BioRender.com, accessed on 5 April 2023).

## Data Availability

Not applicable.

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
