# Peer review of "Adherence to the Mediterranean Diet in Women and Reproductive Health across the Lifespan: A Narrative Review"

_nutrients, 2023, doi:10.3390/nu15092131_

Round 1
Reviewer 1 Report
Aim of this narrative review is to evaluate evidence-based knowledge on the association between mediterranean diet and selected reproductive health outcomes.
The authors addressed an interesting topic where the literature is scarce through a rigorous application of the criteria of narrative review and quality control of the included papers.
For each topic, a paragraph was added commenting on the possible mechanisms through which the Mediterranean diet could positively influence different research areas.
In order to be published, the paper needs a discussion to be added highlighting which characteristics of the Mediterranean diet can give the supposed benefits. For example, the vitamin E in extra virgin olive oil? Or the red wine? Or the high content of whole grains? Or the scarcity of animal protein? Or is it the sum of all this?
Basically, there is a lack of nutrition in this paper
Beware of plagiarism between methods (see attached file)

It can be improved
Author Response
Aim of this narrative review is to evaluate evidence-based knowledge on the association between Mediterranean diet and selected reproductive health outcomes.
The authors addressed an interesting topic where the literature is scarce through a rigorous application of the criteria of narrative review and quality control of the included papers.
For each topic, a paragraph was added commenting on the possible mechanisms through which the Mediterranean diet could positively influence different research areas.
- In order to be published, the paper needs a discussion to be added highlighting which characteristics of the Mediterranean diet can give the supposed benefits. For example, the vitamin E in extra virgin olive oil? Or the red wine? Or the high content of whole grains? Or the scarcity of animal protein? Or is it the sum of all this? Basically, there is a lack of nutrition in this paper.
Response: We would like to clarify that our study focuses on the association between the commonly studied aspect of diet which is adherence to a Mediterranean Diet and thus is primarily centered on nutrition and its potential impact on women’s reproductive health.
To address your comments in the revised version of the manuscript, we have added the section “Beneficial effects of MD on women’s reproductive health: impact of single components or synergistic effect?” in which we have discussed and highlighted the importance of understanding specific characteristics of the Mediterranean diet (MD) components in relation to beneficial effects on women's reproductive health. Furthermore, we have emphasized that it is likely that the overall synergistic effects of the MD dietary pattern rather than isolated components contribute to its beneficial effect on reproductive health (page 17-18, lines 564-593). Moreover, in the subsection “Implications for future research” we have discussed the importance of standardization in the assessment of the MD (page 18, lines 617-627.).
- Beware of plagiarism between methods (see attached file)
Response: We have carefully reviewed the attached file and have taken your comment into consideration. We have revised and paraphrased the relevant sections to further minimize the potential for plagiarism (page 2, line 71-77, 82-83, 85-86). However, we acknowledge that while we have made this effort, the methods used in our study are standardized and have limited options for rephrasing without compromising accuracy and clarity.
Reviewer 2 Report
This is a very well-done review. The review protocol is sound, and the organization is excellent. The topic is valuable and raises gaps in literature with regards to dietary manipulation for various women’s health conditions.
The paper needs editing to address some awkwardness in the English language. Also, some formatting fixes are required.
Line 89: Suggest adding a paragraph discussing quality ratings so that when you describe a low quality study, for example, it is clear why the study is low quality.
Table 1: Change “None study was conducted” To “No study…”
Line 500: Add to potential contributing factor to infertility: Cohen NJ, Yao M, Midya V, et al. Exposure to perfluoroalkyl substances and women's fertility outcomes in a Singaporean population-based preconception cohort. Sci Total Environ. 2023 May 15;873:162267. doi: 10.1016/j.scitotenv.2023.162267.
Line 601: Were the dietary compositions of the MD in the various studies different? If so, mention here that another need is to standardize MD so that more reliable comparisons can be made.
Line 624: Supplementary Materials: Fill in table name. Is there a video? If not, delete reference to video.
The paper needs editing to address some awkwardness in the English language. Also, some formatting fixes are required.
Author Response
This is a very well-done review. The review protocol is sound, and the organization is excellent. The topic is valuable and raises gaps in literature with regards to dietary manipulation for various women’s health conditions.
- The paper needs editing to address some awkwardness in the English language. Also, some formatting fixes are required.
Response: The manuscript has been revised by a native English speaker from a translation agency. During the manuscript resubmission process, we have attached the certificate that it has been done. We also have done some formatting fixes.
- Line 89: Suggest adding a paragraph discussing quality ratings so that when you describe a low quality study, for example, it is clear why the study is low quality.
Response: As recommended we have added the paragraph with a brief explanation of the scoring (page 2-3, lines 88-102). Due to high heterogeneity in the types of included studies, we used 4 different scales to assess the quality of the studies, and each scale took into account slightly different methodological aspects, therefore it was impossible to discuss in detail the rules for awarding points. We would like to emphasize that specific assessment criteria and awarded points are presented in Supplementary Materials (Tables S1-S4), and some limitations of the included studies were also discussed in the text of the manuscript, for example page 9, lines 370-374.
- Table 1: Change “None study was conducted” To “No study…”
Response: Thank you for this comment, we have changed the word (page 6, lines 203).
- Line 500: Add to potential contributing factor to infertility: Cohen NJ, Yao M, Midya V, et al. Exposure to perfluoroalkyl substances and women's fertility outcomes in a Singaporean population-based preconception cohort. Sci Total Environ. 2023 May 15;873:162267.doi: 10.1016/j.scitotenv.2023.162267.
Response: As recommended, we have added a potential contributing factor that may cause female idiopathic infertility (page 13, line 493-497). The reference has also been added (reference no. 97).
- Line 601: Were the dietary compositions of the MD in the various studies different? If so, mention here that another need is to standardize MD so that more reliable comparisons can be made.
Response: As recommended, we have outlined that MD was assessed differently in the included studies and that there is a need to standardize MD assessment (page 18, line 617-627).
- Line 624: Supplementary Materials: Fill in table name. Is there a video? If not, delete reference to video.
Response: We have added the names of supplementary tables (page 19, line 646-651), and the reference to a video was removed.
Round 2
Reviewer 1 Report
The authirs fixed the paper according to my instructions.
It is very surprising that works on menopause are not included. Searching on pubmed I find several papers on menopause and the Mediterranean diet on a wide variety of topics.
I would remove the menopausal period from the paper.
English is ok
Author Response
Thank you for your additional comment. We appreciate your effort to search for related literature on menopause and the Mediterranean diet. We apologize for any confusion we may have caused in our manuscript.
We would like to clarify that we deliberately described only the age of menopause in our manuscript. Menopause is defined as the first day after the last menstrual bleeding and marks the transition between the reproductive and post-reproductive periods. Our search did not find any studies that specifically investigated the association between the Mediterranean diet and the age of onset of menopause. However, we did find several studies on the association between Mediterranean diet adherence and menopausal symptoms as you described, but those studies were conducted in peri- or postmenopausal women. These studies have reported beneficial effects of the Mediterranean diet on hot flashes, night sweats, and other symptoms commonly experienced during the menopausal transition. It is worth noting that some of these studies have been conducted among women of a very wide range of ages (40-75 years) – example: Atalay et al.: Adherence to a Mediterranean diet and cardio-metabolic risk in postmenopausal women by body composition. Asia Pac J Clin Nutr, 2022, 31: 312-319, doi: 10.6133/apjcn.202206_31(2).0017.
We found it hard and controversial to decide whether those symptoms and health problems are a result of menopause or rather it is the result of the age of women or other factors. During the manuscript preparation process, we prepared the paragraph that assessed health problems during the peri- and postmenopausal periods in relation to MD adherence but decided not to include it due to these concerns. We only described the position statement of The European Menopause and Andropause Society (EMAS), which recommends the Mediterranean diet as a dietary pattern that may help to alleviate menopausal symptoms and prevent chronic diseases (page 5, line 207-218). However, based on your comment, in the revised version of the manuscript we have now included this paragraph (page 6-7, line 219-247).
Given the above information, we believe that menopause is a relevant topic to include in our manuscript, especially since it is a time of significant hormonal changes that affect women's reproductive health. However, we ask that you review this new paragraph and decide if you think the studies add value to the overall topic.